# Adherence to the Mediterranean Diet Is Associated with a More Favorable Left Ventricular Geometry in Patients with End-Stage Kidney Disease

**DOI:** 10.3390/jcm11195746

**Published:** 2022-09-28

**Authors:** Dimitra Bacharaki, Ioannis Petrakis, Periklis Kyriazis, Anastasia Markaki, Christos Pleros, Georgios Tsirpanlis, Marios Theodoridis, Olga Balafa, Anastasia Georgoulidou, Eleni Drosataki, Kostas Stylianou

**Affiliations:** 1Nephrology Department, Attikon University Hospital, 12462 Athens, Greece; 2Nephrology Department, University General Hospital of Heraklion, 71500 Iraklio, Greece; 3Division of Nephrology, Beth Israel Deaconess Medical Center, Harvard Medical School, Boston, MA 02215, USA; 4Department of Nutrition and Dietetics, Sciences School of Health Science, Hellenic Mediterranean University, 71410 Heraklion, Greece; 5Nephrology Department, General Hospital of Genimata, 11527 Athens, Greece; 6Department of Nephrology, Democritus University of Thrace, 68150 Alexandroupolis, Greece; 7Nephrology Department, University Hospital of Ioannina, 45500 Ioannina, Greece; 8Nephrology Department, General Hospital of Komotini, 69133 Komotini, Greece

**Keywords:** Mediterranean diet, end-stage kidney disease, cardiac geometry, left ventricular hypertrophy

## Abstract

Introduction. The aim of the study was to examine the impact of adherence to a Mediterranean-style diet (MD) on left ventricular hypertrophy (LVH) and cardiac geometry in chronic kidney disease patients on dialysis (CKD-5D), given the high prevalence of cardiovascular morbidity in this population. Methods. *n* = 127 (77 men and 50 women) CKD-5D patients (69 on hemodialysis and 58 on peritoneal dialysis) with a mean age of 62 ± 15 years were studied. An MD adherence score (MDS) (range 0–55, 55 representing maximal adherence) was estimated with a validated method. Echocardiographic LVH was defined by LV mass index (LVMI) > 95 g/m^2^ in women and >115 g/m^2^ in men. Based on LVMI and relative wall thickness (RWT), four LV geometric patterns were defined: normal (normal LVMI and RWT), concentric remodeling (normal LVMI and increased RWT > 0.42), eccentric LVH (increased LVMI and normal RWT), and concentric LVH (increased LVMI and RWT). Results. Patients with LVH (*n* = 81) as compared to patients with no LVH (*n* = 46) were older in age (66 ± 13 vs. 55 ± 16 years; *p* < 0.001) had lower MDS (24 ± 2.7 vs. 25 ± 4.3; *p* < 0.05) and higher malnutrition-inflammation score (5.0 ± 2.7 vs. 3.9 ± 1.9; *p* < 0.05), body mass index (27.5 ± 4.9 vs. 24.1 ± 3.5 kg/m^2^; *p* < 0.001), prevalence of diabetes (79% vs. 20%; *p* < 0.05), coronary artery disease (78% vs. 20%; *p* < 0.05) and peripheral vascular disease (78% vs. 20%; *p* < 0.01). In a multivariate logistic regression analysis adjusted for all factors mentioned above, each 1-point greater MDS was associated with 18% lower odds of having LVH (OR = 0.82, 95% CI: 0.69–0.98; *p* < 0.05). MDS was inversely related to LVMI (r = −0.273; *p* = 0.02), and in a multiple linear regression model (where LVMI was analyzed as a continuous variable), MDS emerged as a significant (Β = −2.217; *p* < 0.01) independent predictor of LVH. Considering LV geometry, there was a progressive decrease in MDS from the normal group (25.0 ± 3.7) to concentric remodeling (25.8 ± 3.0), eccentric (24.0 ± 2.8), and then concentric (23.6 ± 2.7) group (*p* < 0.05 for the trend). Conclusions. The greater adherence to an MD is associated with lesser LVH, an important cardiovascular disease risk factor; MD preserves normal cardiac geometry and may confer protection against future cardiac dysfunction in dialysis patients.

## 1. Introduction

There is an established relationship between renal function deterioration and the occurrence of major cardiovascular events (MACE), especially in chronic kidney disease stage-five dialysis (CKD-5D) patients [1,2,3]. Diabetes, arterial hypertension, obesity, and smoking constitute traditional risk factors for MACE in CKD-5D patients [4,5,6]. Dyslipidaemia, anemia, components of mineral metabolism such as serum phosphate, FGF23 levels, active vitamin D metabolites, and hyperparathyroidism have been implicated as MACE contributors in CKD-5D. However, the lack of specifically designed randomized clinical trials for the CKD-5D population weakens the association of the aforementioned cardiovascular risk factors with MACE [7,8,9,10,11,12,13,14].

In his pivotal research, Ganau et al. [15] described three different LVH patterns in patients with essential hypertension. LV mass indexed by height (LVMI g/m^2^) and relative wall thickness of the left ventricle (RWT) were utilized to classify patients as having concentric remodeling (cLVR), eccentric LVH (eLVH), and concentric LVH (cLVH). More specifically, in the CKD-5D population, cLVR is defined as increased RWT (RWT > 0.42) and normal LVMI (LVMI ≤ 115 g/m^2^ in men or ≤95 g/m^2^ in women). eLVH is defined as normal RWT (RWT ≤ 0.42) and increased LVMI. Finally, cLVH is defined by both increased RWT and LVMI [16].

CKD patients frequently present with cLVH and eLVH [17]. Both these cardiac geometry patterns were associated with an increased cardiovascular risk [17]. The presence of eLVH and cLVH was associated with all-cause mortality and also with renal function deterioration [16,18]. LVH is extremely common in CKD-5D patients [19], and eLVH was associated with CVD in a CKD-5D cohort in the long term [20]. Therefore, there seems to be an apparent association between LVH and CVD in the dialysis population.

Diets with favorable cardiovascular metabolic profiles [21], such as the Mediterranean diet, have been associated with favorable effects on LV structure and function [22,23,24]. Mediterranean Diet is a powerful weapon addressing simultaneously all adverse pathophysiologic mechanisms of renal failure [25], but vegetable-based diets have always been a concern among patients and Nephrologists because of the fear of increased potassium [26].

This study explores the association of adherence to the Mediterranean diet (MD) with LVH and cardiac geometry patterns in a cohort of CKD-5D patients.

## 2. Materials and Methods

### 2.1. Study Population

The study was performed at nine dialysis units in Greece. All study participants were of Caucasian ancestry. Patients were included when they had been on dialysis for at least 3 months and were 18 years or older. Exclusion criteria were recent major surgery, malignant disease, concurrent inflammatory illness, life expectancies of less than 1 year, cognitive impairment, and unwillingness to participate (see flow diagram in Appendix A). Enrolled hemodialysis (HD) patients were treated 3 times per week for a minimum of 4 h using bicarbonate dialysate and aiming for a minimum target KT/V of 1.2 as recommended [27]. Peritoneal dialysis (PD) patients were on a standard continuous ambulatory PD program (4–5 exchanges per day) or on automated peritoneal dialysis prescribed high-quality goal-directed peritoneal dialysis, as described [28]. Patients’ recruitment into the study lasted from November 2019 to March 2020. The study was performed in strict accordance with the ethical guidelines of the Helsinki Declaration and was approved by the Ethical Scientific Committee of the participating centers (decision 11795/09-10-2019). All study participants provided written informed consent.

### 2.2. Clinical Data

Each patient’s medical chart was reviewed, extracting data pertaining to a history of peripheral vascular disease (PVD), coronary artery disease (CAD), stroke, diabetes, arterial hypertension, and prescribed antihypertensive medications. Baseline PVD was defined as the development of symptoms (intermittent claudication), therapeutic interventions (revascularisation and amputation), and artery stenosis > 60% in imaging studies. Coronary artery disease (CAD) was defined as a medical history of myocardial infarction, angina pectoris, percutaneous coronary intervention, and coronary artery bypass surgery. The rationale for including patients with angina pectoris as patients with CAD is the following: CKD-5D patients are less likely to receive an invasive strategy for diagnosing and treating CAD [29,30]. Furthermore, an increase in biomarkers of myocardial injury is less likely to be indicative of NSTEMI. The presence of chronic myocardial injury, which is common in CKD-5D patients, might as well result in an increase in myocardial injury biomarkers [29]. Therefore, the presence of angina pectoris was considered a CAD analog that might affect myocardial geometry. Arterial hypertension was defined as systolic blood pressure >140 mmHg or the use of antihypertensive treatment up to 6 months before enrolment. Clinical information collected at baseline for each patient also included age, dialysis vintage, body mass index (BMI), and smoking habits

### 2.3. Mediterranean Diet Score

Adherence to the MD diet was assessed by the Med Diet Score (MDS; Table 1), as described previously by Panagiotakos et al. [31]. Briefly, a dietician was assigned to interrogate patients about the weekly consumption of eleven food groups: non-refined cereals, fruit, vegetables, legumes, potatoes, fish, red meat and products, poultry, full-fat dairy products, olive oil, and alcohol intake. For the consumption of items presumed to be close to Mediterranean dietary patterns (non-refined cereals, fruits, vegetables, legumes, olive oil, fish, and potatoes), scores of 0–5 reflect increased consumption in positive correlation, while for the consumption of foods presumed to be away from this pattern (red meat and products, poultry and full-fat dairy products) scores on a reverse scale were assigned. The total score is given a number from 0 to 55, the highest reflecting maximum Med diet adherence. To further explore the relationship between individual food categories and important CVD risk factors, a separate analysis was performed involving 3 groups: (A) the “avoid foods” group comprising the 3 ingredients which were ascribed a negative value by incremental consumption (red meat and products, poultry, full-fat dairy products), (B) the “recommended foods’ group comprising the 7 ingredients which gained a positive value by incremental consumption (Non-refined cereals, potatoes, fruits, vegetables, legumes, fish, olive oil) and (C) the “fruits, vegetables, legumes-FVL” group.

### 2.4. Echocardiography

Echocardiographic studies were performed before the midweek dialysis session for the HD patients and the day of the scheduled visit for the PD patients. LV dimensions were determined from a two-dimensional, guided M-mode tracing as described [32]. LV posterior wall thickness (PWT), interventricular septum thickness (IVS), and diastolic and systolic internal diameters (LVDD and LVSD) were all measured just below the tip of the mitral valve leaflets at the peak of the R wave on the electrocardiogram. LV mass was then calculated according to the Devereux formula: LV Mass (g) = 0.8[1.04[([LVEDD + IVSd + PWd]3 − LVEDD3)]] + 0.6, where LVEDD, IVSd, and PWd represent LV, interventricular septal, and posterior wall thickness in diastole [33]. For BSA, the Mosteller’s formula was employed: BSA (m^2^) = Square root ((Height (cm) × Weight (kg))/3600).

Echocardiographic LVH was defined by LV mass index (LVMI) > 95 g/m^2^ in women and >115 g/m^2^ in men [33,34]. Relative wall thickness (RWT) was calculated by the formula RWT = 2 × PWd/LVEDD. Based on LVMI and relative wall thickness (RWT), four LV geometric patterns were defined: normal (normal LVMI and RWT), concentric remodeling (normal LVMI and increased RWT > 0.42), eccentric LVH (increased LVMI and normal RWT), and concentric LVH (increased LVMI and RWT) [33].

### 2.5. Malnutrition-Inflammation Score

Malnutrition-inflammation score (MIS), as described by Kalantar-Zadeh et al. [35], was calculated for all patients. MIS consists of 4 domains, assessing patients’ medical history, physical examination, BMI and laboratory parameters, and 10 components. The total score deriving from all MIS components ranges from 0 to 30, with higher scores reflecting an increased risk of malnutrition and inflammation

### 2.6. Laboratory Measurements

For laboratory testing, blood samples were collected before the mid-week dialysis session for the HD patients and on the morning of the scheduled visit for the PD patients. Hemoglobin (Hb), serum albumin (sAlb), creatinine, total cholesterol, high-density lipoprotein (HDL) and low-density lipoprotein (LDL) cholesterol, triglycerides, serum calcium (sCa), serum magnesium (sMg), serum Potassium (sK) and serum phosphorous (sP) were measured routinely, at least once monthly and serum parathormone (PTH) at least quarterly, using standard laboratory techniques. The average of the last 3-months’ laboratory data was used for analysis.

### 2.7. Statistical Analysis

For all statistical analyses, the SPSS/PC 20 statistical package (IBM, Chicago, IL, USA) was used. Normally distributed variables were expressed as mean ± SD, and non-normally distributed variables were expressed as median (interquartile range). The Chi-square test was used for the analysis of categorical variables and Student’s *t*-test for continuous variables, as appropriate. MDS was examined both as a continuous and a dichotomous variable, in the latter case, categorized according to its median. Stepwise multiple logistic regression analysis was performed to determine significant factors associated with LVH, according to the diagnostic criteria mentioned above. Univariate and multivariate regression analyses were used to test the associations between variables. Statistical significance was set at the level of *p* < 0.05 (two-sided).

## 3. Results

### 3.1. Study Population

The study cohort consisted of 127 patients with a mean age of 62 ± 15 (range 20–91) years, as shown in Table 2. Sixty-nine patients (48 men and 21 women, mean age 62 ± 16 years) underwent HD, and 58 patients (29 men and 29 women, mean age 62 ± 14 years) were treated with PD. Diabetes, PVD and CAD were detected in 26.8%, 37% and 28.3% of patients, respectively, There were 96 (75.6%) hypertensive patients and most of them 88 (91.7) were on antihypertensive drugs [b-blockers, *n* = 67; calcium channel blockers, *n* = 45; angiotensin-converting enzyme inhibitors/angiotensin receptor blockers, *n* = 48; diuretics, *n* = 31. There were 25 (22.1%) current smokers. Dialysis vintage was shorter in PD than HD patients (37 (11.5–65) vs. 51.5 (31–102) months; *p* < 0.05). There was no significant difference between HD and PD patients in terms of LVH prevalence (58% vs. 42%) and LVMI (116 ± 24 vs. 111 ± 33 g/m^2^).

### 3.2. Patients with and without Left Ventricular Hypertrophy (LVH)

Eighty-one patients (63.8%) had LVH, as defined in the Materials and Methods section, and 46 (36.2%) patients were without LVH. The general characteristics of the two groups are shown in Table 2. Patients with LVH were older, had higher BMI, pulse pressure (PP), sCa and MIS, lower sAlb, sMg, and MDS, showed an increased prevalence of diabetes, PVD, and CAD, and made greater use of CCBs. The two groups did not differ significantly from each other in terms of sex, dialysis mode, dialysis vintage, use of calcimimetics, blood pressure, hypertension, smoking, creatinine, hemoglobin, lipid profile, sP, potassium, and parathormone.

### 3.3. Predictors of Left Ventricular Hypertrophy (LVH)

Factors associated with LVH were determined with univariate logistic regression analysis (Table 3). Controlling for dialysis mode, sex, and factors with the statistically significant association on univariate analysis (age, PVD, CAD, diabetes, BMI, PP, sAlb, sMg, MIS), backward selection logistic analysis indicated that each increase in MDS by 1 point was associated with 18% lower odds of having LVH (OR = 0.82, 95% CI: 0.68–0.99; *p* = 0.037). Except for MDS, CAD, sex, BMI, pulse pressure, sMg, and MIS also emerged as independent predictors of LVH. Notably, women had a 7.3-fold increased risk for LVH. In this regard, 19 out of 21 women on HD had developed LVH.

### 3.4. Determinants of Left Ventricular Mass Index (LVMI)

MDS was associated with LVMI (r = −0.273; *p* = 0.002; Figure 1). In addition, as shown in Table 2, LVMI was positively associated with age, sex, BMI, PVD, hypertension, systolic blood pressure, pulse pressure, smoking, MIS, and the use of CCBs and b-blockers, whereas it was inversely related to sMg. The results of a forward stepwise regression analysis (Table 4), where variables significant in univariate analysis were included, showed that MDS, age, smoking, and pulse pressure were independently associated with LVH as indicated by LVMI. Examination of the standardized (beta = −0.243) and partial correlation (r = −0.265) coefficients indicated that MDS was the strongest determinant of LVH, explaining almost 9% of the variability of LVMI. Together, these four variables explained 23% of the variance in LVMI.

### 3.5. ΜDS as Major Determinant of Left Ventricular Geometry Patterns

Considering LV geometry among all participants, eccentric hypertrophy was the most prevalent LV geometric pattern (33.9%; *n* = 43), followed by concentric hypertrophy (26%; *n* = 33), normal geometry (23.6% *n* = 30) and concentric remodeling (16.5%; *n* = 21) (Figure 2).

There was a progressive decrease (*p* = 0.032 for the trend) in MDS from the normal group (24.9 ± 3.5) to concentric remodeling (25.9 ± 2.9), eccentric (24.1 ± 2.8), and then to the concentric (23.6 ± 2.7) hypertrophy group (Figure 3).

### 3.6. Determinants of MDS

Given that MDS substantially impacted LVH and LV geometrical patterns, next, we examined factors associated with MDS. As mentioned in the Methods, MDS was analyzed both as a dichotomous variable (dichotomized into above (high) and below (low)), with the MDS median of 24 as a continuous variable. According to data shown in Table 5, patients in the high MDS group had higher sAlb, HDL cholesterol, sMg, and lower systolic and diastolic blood pressure, LDL cholesterol, sCa, MIS, and prevalence of PVD. Also, according to Pearson correlation analysis (Table 5), MDS was associated directly with sMg, sAlb, and HDL cholesterol and inversely with PVD, blood pressure, LDL, sCa, and MIS in our patients. Interestingly, serum potassium was not associated with MD adherence.

### 3.7. MDS as a Risk Indicator for CVD

To provide further proof of the beneficial effect of adherence to MD, especially in the context of prevention of CVD, we examined the association of MDS categories (quartiles) with cardiac structure and LVH phenotypes, co-morbidities, and antihypertensive drugs (Table 6). Participants with the highest levels of adherence to the MD (Q4) had a significantly lower risk of developing LVH (37.5% vs. 71.4% in Q1; *p* = 0.009) and abnormal LV geometry (eLVH: 18.8% in Q4 vs. 35.7% in Q1 and cLVH: 12.5% in Q4 vs. 32.1% in Q1; *p* = 0.020). In addition, patients in the highest quartile (Q4) had the lowest prevalence of PVD (18.8% vs. 43.2% in Q1–Q3 combined; *p* = 0.013) and stroke (3.1% vs. 18.9% in Q1–Q3 combined; *p* = 0.030). Finally, considering that a higher MDS was associated with the lowest rate of b-blockers use, greater adherence to MD may even reduce the demand for antihypertensive drugs, particularly b-blockers, which are beneficial in the treatment of CVD. Finally, BMI in the higher quartile (Q4) of MDS distribution (25.2 ± 3.6 Kg/m^2^) was lower as compared to that of Q1–Q3 combined (26.6 ± 5.0 Kg/m^2^), without however reaching statistical significance (*p* = 0.151). Taken together, all these data suggest that MDS could help identify CKD-5D patients at greater CV risk.

### 3.8. Associations of Risk Factors for CVD with Med Diet Food Categories

To further explore the relationship between individual food categories and important CVD risk factors, a separate analysis was performed involving 3 groups: (A) the “avoid foods” group comprising red meat and products, poultry, full-fat dairy products, (B) the “recommended foods’ group comprising 7 ingredients; non-refined cereals, potatoes, fruits, vegetables, legumes, fish, olive oil and (C) the “fruits, vegetables, legumes-FVL” group. As shown in Table 7, there was a significant decrease in sMg levels and systolic blood pressure by avoiding foods comprising group A. Consumption of recommended foods (group B) was associated with a lower LVMI, less distortion of left ventricular geometry, lower prevalence of LVH and PVD, lower LDL and higher sMg and albumin levels. Apart from the correlations mentioned above, a subset of recommended foods included in group C (FVL) was additionally associated with a lower prevalence of stroke and lower MIS.

## 4. Discussion

The main finding of the present study is that higher adherence to MD in our CKD stage 5D patients was associated with a lower prevalence of LVH. This association remained statistically significant upon adjustment for sex, age, PVD, CAD, diabetes, BMI, PP, sMg, sAlb, and MIS, factors that might confound the observed association. We showed that each 1-point greater MDS was associated with 18% lower odds of having LVH. In this regard, there was an average 2.22 g/m^2^ decrease in LVMI for every 1-point increase in the MDS. Equally notable, higher adherence to the Mediterranean diet was linked with a lower prevalence of the most serious LV geometric abnormalities (eccentric and concentric hypertrophy) in our dialysis population.

Rees et al., in a meta-analysis of the available studies on the effects of MD in primary or secondary cardiovascular prevention, showed no certain association regarding cardiovascular endpoints [36]. Nevertheless, a targeted examination of left ventricular parameters with cardiac MRI in 4497 healthy volunteers in regard to the Mediterranean diet showed no adverse cardiac remodeling and found no association between LVH and MD patterns [22]. Northern Manhattan Study participants exhibited an inverse association between Mediterranean diet adherence and LV mass [23]. Furthermore, in a cross-sectional study of 438 healthy subjects, Maugeri et al. [37] associated a decrease in cLVH with MD. Mediterranean diet comprises low salt and low phosphorus intake compared with western style diets. Increased phosphate and sodium load have been associated with left ventricular hypertrophy in the dialysis population [38,39]. To our knowledge, available evidence regarding the Mediterranean diet and LVH is rare. Given the adverse effect of less favorable LVH patterns in CKD and dialysis patients [16,17,18,19,20], diet style per se could serve as an intervention that could reduce CKD-associated cardiovascular mortality.

Hypertension in patients with CKD-5D has been shown to be associated with increased mortality [40]. Malnutrition inflammation score depicts the inherent hypercatabolic and inflammation states which are present in a subset of the dialysis population [35]. An increase in MIS has been associated with mortality in CKD-5D patients [41,42]. Decreased serum magnesium is also associated with increased mortality in dialysis patients [43,44,45]. In our study, increased adherence to the Mediterranean diet resulted in decreased systolic blood pressure, decreased malnutrition inflammation score, increased serum albumin, and increased serum magnesium. All these associations might collectively contribute to a favorable cardiac geometry in dialysis patients with a Mediterranean-style nutrition pattern.

From our data, it seems apparent that the consumption of specific food categories, such as vegetables and legumes, was associated with a more favorable cardiac remodeling pattern. In accordance with our observations, Hu et al. showed that consumption of legumes and vegetables in the long term was associated with reduced CKD progression and cardiovascular mortality [46]. Therefore, it might be that specific nutritional components within a dietetic style might contribute to a greater or lesser extent to a specific pattern of cardiac remodeling.

Diet-associated hyperkalemia is a major concern in CKD-5D patients. MD constituents are rich in potassium. Interestingly, adherence to MD was not associated with hyperkalemia in this cohort of dialysis patients, thus forming a tempting argument for its safe implementation in dialysis patients.

The relatively small number of patients and the focus on subjects of a single ancestry are weaknesses of the present study. Furthermore, the lack of randomization and the presence of a dietetic intervention underpower the significance of our results. Defining angina pectoris as a CAD analog might result in the over-diagnosis of CAD within our patient cohort. Echocardiographic evaluation was not based on the newest recommendation on chamber quantification in 2015 [47]. To avoid a measurement/classification bias, given the operator variability in echocardiographic findings, we preserved the measurement definitions as described in the 2005 guideline on chamber quantification [33]. This fact and the absence of an additional, more precise method, such as cardiac MRI, to confirm the results obtained through echocardiography should be added to the weaknesses of the present study.

Conclusively we show an association between a Mediterranean-style diet with a favorable cardiac remodeling pattern in a cohort of hemodialysis and peritoneal dialysis patients. This poses the notion that in CKD-5D patients, nutritional interventions might have a therapeutic role as well.

## Figures and Tables

**Figure 1 jcm-11-05746-f001:**
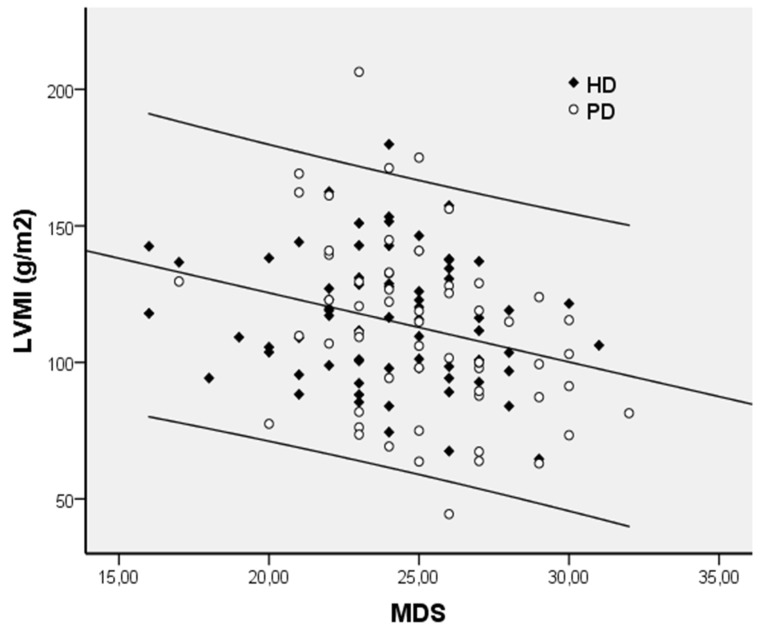
The correlation between left ventricular mass index (LVMI) and Mediterranean diet score (MDS); HD, Hemodialysis; PD, Peritoneal Dialysis.

**Figure 2 jcm-11-05746-f002:**
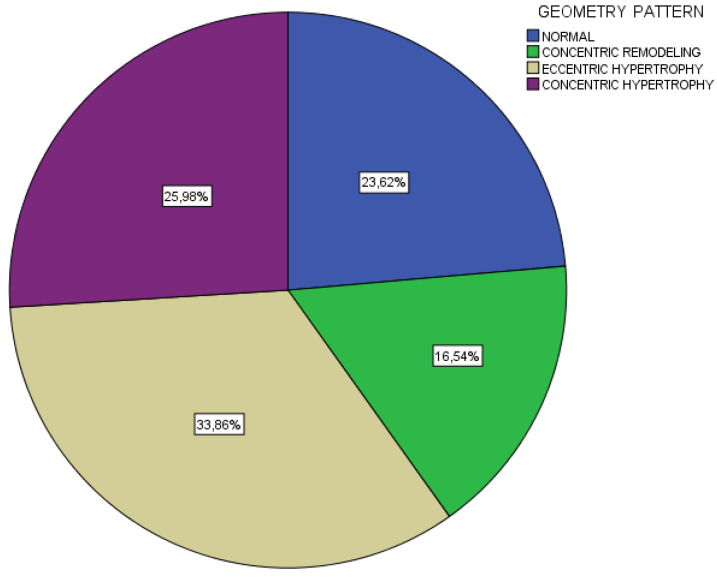
Distribution of cardiac geometry patterns (*n* = 127 chronic kidney disease patients on dialysis).

**Figure 3 jcm-11-05746-f003:**
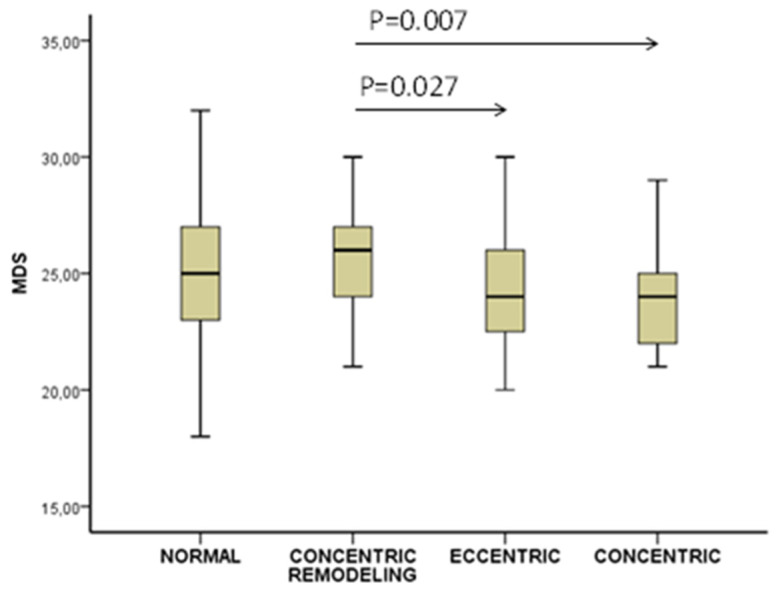
Mediterranean diet score (MDS) by left ventricular geometry patterns.

**Table 1 jcm-11-05746-t001:** The Mediterranean Diet Score.

How Often Do You Consume (Servings/Week or Otherwise Stated)	Frequency of Consumption					
Non-Refined Cereals (Whole Grain Bread, Pasta, Rice, etc.)	Never	1–6	7–12	13–18	19–31	>32
	0	1	2	3	4	5
Potatoes	Never	1–4	5–8	9–12	13–18	>18
	0	1	2	3	4	5
Fruits	Never	1–4	5–8	9–15	16–21	>22
	0	1	2	3	4	5
Vegetables	Never	1–6	7–12	13–20	21–32	>33
	0	1	2	3	4	5
Legumes	Never	<1	1–2	3–4	5–6	>6
	0	1	2	3	4	5
Fish	Never	<1	1–2	3–4	5–6	>6
	0	1	2	3	4	5
Red meat and products	≤1	2–3	4–5	6–7	8–10	>10
	5	4	3	2	1	0
Poultry	≤3	4–5	5–6	7–8	9–10	>10
	5	4	3	2	1	0
Full fat dairy products (cheese, yogurt, milk)	≤10	11–15	16–20	21–28	29–30	>30
	5	4	3	2	1	0
Use of olive oil in cooking (times/week)	Never	Rare	<1	1–3	3–5	Daily
	0	1	2	3	4	5
Alcoholic beverages (mL/day, 100 mL = 12 g ethanol)	<300	300	400	500	600	>700 or 0
	5	4	3	2	1	0

**Table 2 jcm-11-05746-t002:** Clinical and epidemiological characteristics of the study population by left ventricular hypertrophy (LVH).

Characteristic	All Patients (*n* = 127)	LVH (*n* = 81)	NO LVH (*n* = 46)	*p*-Value	r	*p*-Value
Age (year)	62 ± 15	66 ± 13	55 ± 16	**0.000**	0.270	**0.002**
Sex (male/females)	77/50	31/15	46/35	0.240	−0.187	**0.035**
Dialysis mode [(HD = 1/PD = 2) (*n*)]	69/58	22/24	47/34	0.267	−0.082	0.358
Dialysis vintage (months)	36 (15–60)	36 (17–58)	35 (12–68)	0.425	−0.129	0.151
Diabetes mellitus (%)	26.8	33.3	15.2	**0.027**	0.093	0.278
Peripheral vascular disease (%)	37	44.4	23.9	**0.021**	0.191	**0.031**
Coronary artery disease (%)	28.3	34.6	17.4	**0.039**	0.158	0.076
Hypertension (%)	75.6	79	69.6	0.234	0.178	**0.046**
Stroke (%)	15	17.3	10.9	0.330	0.011	0.905
Smoking (%)	22.1	26.4	14.6	0.148	0.255	**0.006**
Body mass index (Kg/m^2^)	26.2 ± 4.7	27.5 ± 4.9	24.1 ± 3.5	**0.000**	0.230	**0.009**
SBP (mmHg)	133 ± 19	135 ± 19	129 ± 28	0.086	0.259	**0.03**
DBP (mmHg)	76 ± 14	76 ± 13	776 ± 15	0.891	0.019	0.833
Pulse pressure (mmHg)	57 ± 15	59 ± 14	53 ± 15	**0.009**	0.321	**0.000**
LVMI index (g/m^2^)	114.2 ± 28.1	129.5 ± 21.2	87.2 ± 15.4	0.000	−	−
ACEi/ARBs (%)	42.5	45.7	37	0.339	0.162	0.069
B-BLOCKERS (%)	67.5	71.6	60	0.183	0.182	**0.042**
CCB (%)	40.5	46.9	28.9	**0.048**	0.238	**0.007**
DIURETICS (%)	28.3	28.4	28.3	0.987	0.003	0.978
CALCIMIMETICS (%)	32.3	27.2	41.3	0.075	−0.156	0.080
**Nutritional and biochemical**						
Albumin (g/dL)	3.6 ± 0.5	3.6 ± 0.5	3.8 ± 0.4	**0.011**	−0.119	0.183
Creatinine (mg/dL)	8.7 ± 3.0	8.3 ± 2.8	9.4 ± 2.3	0.054	−0.051	0.569
Hemoglobin (g/dL)	11.7 (11.1–12.4)	11.7 (11.2–12.2)	11.7 (11.0–12.6)	0.887	−0.103	0.250
Total Cholesterol (mg/dL)	159 ± 35	159 ± 35	159 ± 36	0.969	−0.069	0.441
HDL (mg/dL)	43 ± 11	42 ± 11	46 ± 11	0.093	−0.158	0.077
LDL (mg/dL)	85 ± 28	85 ± 29	86 ± 28	0.832	−0.058	0.519
Triglycerides (mg/dL)	140 (103–184)	133 (105–191)	140 (100–178)	0.477	0.002	0.914
Calcium (mg/dL)	9.2 ± 0.6	9.3 ± 0.6	9.1 ± 0.6	**0.034**	0.173	0.052
Magnesium (mg/dL)	2.2 ± 0.4	2.1 ± 0.3	2.4 ± 0.4	**0.000**	−0.219	**0.014**
Potassium (mmol/L)	4.72 ± 0.68	4.83 ± 0.67	4.67 ± 0.70	0.208	−0.011	0.902
Phosphorus (mg/dL)	5.0 ± 1.3	5.0 ± 0.1.3	5.0 ± 1.3	0.958	0.010	0.914
Parathormone (pg/mL)	262 (144–404)	256 (148–408)	275 (143–398)	0.857	0.081	0.365
MIS	4,56 ± 2.47	5.0 ± 2.7	3.9 ± 1.9	**0.017**	0.237	**0.007**
MDS	24.4 ± 3	24 ± 2.7	25 ± 3.4	**0.016**	−0.273	**0.002**

Values are expressed as means ± SDs or medians (interquartile ranges). Significant differences or correlations are shown with bold numbers. HD, hemodialysis; PD, peritoneal dialysis; SBP, systolic blood pressure; DBP, diastolic blood pressure; LVMI, left ventricular mass index; ACEi, angiotensin-converting enzymes inhibitors; ARB, angiotensin II receptors blockers, CCB, calcium channel blockers; HDL high-density lipoprotein, LDL, low-density lipoprotein; MIS, malnutrition-inflammation score, MDS, Mediterranean diet score. r, Pearson correlation coefficient between baseline characteristics and LVMI.

**Table 3 jcm-11-05746-t003:** Risk factors for left ventricular hypertrophy as assessed by multiple logistic regression analysis (*n* = 127).

Parameter	Univariate		Multivariable	
	OR (95%CI)	*p*	OR (95%CI)	*p*
Age (↑ 1 year)	1.05 (1.02–1.08)	0.000	-	-
PVD (presence)	2.55 (2.14–5.70)	0.023	-	-
CAD (presence)	2.51 (1.03–6.11	0.043	8.81(2.25–34.49)	0.002
Diabetes (presence)	2.79 (1.10–7.04	0.030	-	-
ΒΜΙ (↑ 1 Kg/m^2^)	1.21 (1.09–1.35)	0.000	1.33 (1.14–1.55)	0.000
PP (↑ 1 mmHg)	1.04 (1.01–1.06)	0.011	1.05 (1.01–1.09)	0.013
sAlb (↑ 1 g/dL)	0.38 (0.18–0.82)	0.013	-	-
MIS (↑ 1 point)	1.28 (1.03–1.47)	0.020	1.28 (1.01–1.61)	0.039
sMg (↑ mg/dL)	0.13 (0.04–0.41)	0.000	0.08 (0.02–0.38)	0.002
MDS (↑ 1 point)	0.85 (0.75–0.97)	0.018	0.82 (0.68–0.99)	0.037
Sex (male = 1/female = 2)	1.57 (0.74–3.35)	0.247	7.29 (2.11–25.16)	0.002.

PVD, peripheral vascular disease; CAD, coronary artery disease; BMI, body mass index; PP, pulse pressure; sAlb, serum albumin; MIS, malnutrition-inflammation score; sMg, serum magnesium; MDS, Mediterranean diet score; ↑, increase per.

**Table 4 jcm-11-05746-t004:** Predictors of the left ventricular mass index (LVMI) assessed through multiple regression analysis.

Parameter	B	Standard Error	Standard B	*p*	Partial r	Adjusted R^2^ Change (Cumulative Variability)
(Constant)	114,449	23,85		0		
MDS	−2.217	0.778	−0.243	0.005	−0.265	0.089 (0.089)
AGE	0.432	0.156	0.235	0.007	0.258	0.06 (0.149)
SMOKING	16.023	5.772	0.234	0.006	0.258	0.046 (0.195)
PP	0.413	0.166	0.214	0.014	0.233	0.036 (0.231)

Standard beta, standardized regression coefficients; r, correlation coefficient; MDS, Mediterranean-diet-score; PP, pulse pressure.

**Table 5 jcm-11-05746-t005:** Factors associated with MDS.

Factor	Low MDS ≤ 24 (*n* = 63)	High MDS > 24 (*n* = 64)	*p*	r	*p*
PVD (%)	46	28.1	0.037	−0.201	0.024
SBP (mmHg)	137 ± 17	129 ± 20	0.013	−0.283	0.001
DBP (mmHg)	78 ± 14	73 ± 13	0.044	−0.233	0.008
HDL (mg/dL)	41 ± 10	45 ± 11	0.032	0.207	0.020
LDL (mg/dL)	90 ± 32	81 ± 23	0.046	−0.190	0.033
sMg (mg/dL)	2.2 ± 0.3	2.3 ± 0.4	0.048	0.240	0.007
sCa (mg/dL)	9.3 ± 0.5	9.1 ± 0.6	0.046	−0.189	0.034
sAlb (g/dL)	3.5 ± 0.5	3.7 ± 0.5	0.023	0.228	0.010
MIS (points)	4.7 ± 2.7	4.4 ± 2.3	0.398	−0.179	0.044
K (mmol/L)	4.75 ± 0.68	4.70 ± 0.69	0.696	−0.040	0.696

r, Pearson correlation coefficient; PVD, peripheral vascular disease, SBP, systolic blood pressure; DBP, diastolic blood pressure; HDL high-density lipoprotein, LDL, low-density lipoprotein; sMg, serum magnesium; sCa, serum calcium; sAlb, serum albumin; MIS, malnutrition-inflammation score; K, potassium.

**Table 6 jcm-11-05746-t006:** Cardiac measurements, co-morbidities, antihypertensive drugs, and BMI according to the quartiles of MDS.

Variables	Q1 (≤22) *n* = 28	Q2 (23–24) *n* = 37	Q3 (25–26) *n* = 30	Q4 (≥27) *n* = 32	*p* for Trend
Cardiac measures					
A. LVH					0.005
absent	27.6 (8)	27.8 (10)	26.7 (8)	62.5 (20)	
present	71.4 (20)	73.0 (27)	73.3 (22)	37.5 (12)	
B. Geometry pattern					0.041
normal	24.1 (7)	19.4 (7)	16.7 (5)	34.4 (11)	
cLVR	6.9 (2)	11.1 (4)	13.3 (4)	34.4 (11)	
eLVH	35.7 (10)	37.8 (14)	43.3 (13)	18.8 (6)	
cLVH	32.1 (9)	33.3 (12)	26.7 (8)	12.5 (4)	
Co-morbidities					
PVD	39.3 (11)	51.4 (19)	36.7 (11)	18.8 (6)	0.048
CAD	14.3 (4)	32.4 (12)	36.7 (11)	28.1 (9)	0.256
STROKE	14.3 (4)	27 (10)	13.3 (4)	3.1 (1)	0.050
Anti-HTN drugs					
ACEI/ARB	57.1 (16)	37.8 (14)	26.7 (8)	50 (16)	0.086
B-BLOCKERS	78.6 (22)	64.9 (24)	80 (24)	48.4 (15)	0.030
CCB	46.4 (13)	37.8 (14)	43.3 (13)	35.5 (11)	0.816
BMI (mean ± SD)	26 ± 5.0	26.9 ± 5.1	26.6 ± 5.0	25.2 ± 3.6	0.502

Values are expressed as percentages and numbers (in parentheses). LVH, left ventricular hypertrophy; cLVR, concentric LV remodeling; eLVH, eccentric LVH, cLVH; concentric LVH; PVD, peripheral vascular disease; CAD, coronary artery disease; HTN, hypertensive; ACEI, angiotensin-converting enzymes inhibitors; ARB, angiotensin II receptors blockers; CCB, calcium channel blockers.

**Table 7 jcm-11-05746-t007:** Associations of CVD risk factors with Mediterranean Diet food categories.

	GROUP A ‘Avoid Foods’		GROUP B ‘Recommended Foods’		GROUP C ‘FVL’	
Factor	r	*p*	r	*p*	r	*p*
LVMI	−0.210	0.210	−0.213	**0.016**	−0.235	**0.008**
* Cardiac geometry	0.079	0.375	−0.254	**0.004**	−.0329	**0.000**
LVH	−0.019	0.832	−0.235	**0.008**	−0.236	**0.008**
PVD	−0.075	0.405	−0.203	**0.022**	−0.115	0.198
Stroke	−0.077	0.389	−0.130	0.145	−0.175	**0.049**
SBP	−0.191	**0.032**	−0.143	0.110	−0.137	0.125
LDL	−0.101	0.260	−0.209	**0.019**	0.155	0.083
sMg	−0.193	**0.030**	0.396	**0.000**	0.395	**0.000**
MIS	0.001	0.988	−0.128	0.152	−0.176	**0.047**
Albumin	0.001	0.988	0.259	**0.003**	0.188	**0.035**

r, Pearson correlation coefficient; FVL, fruits, vegetables, legumes; LVMI, left ventricular mass index; LVH, left ventricular hypertrophy; PVD, peripheral vascular disease; SBP, systolic blood pressure; LDL, low-density lipoprotein; sMg, serum magnesium; MIS, malnutrition-inflammation score. * Normal pattern = 1, concentric remodeling = 2, eccentric hypertrophy = 3, concentric hypertrophy = 4. Significant correlations are shown with bold numbers.

## Data Availability

The data that support the findings of this study are available from the corresponding author upon reasonable request.

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
