# Peer review of "Adherence to the Mediterranean Diet Is Associated with a More Favorable Left Ventricular Geometry in Patients with End-Stage Kidney Disease"

_jcm, 2022, doi:10.3390/jcm11195746_

Round 1

Reviewer 1 Report

Bacharaki et al. investigated the association between a Mediterranean diet and different patterns of left ventricular hypertrophy (LVH) in a cohort of chronic kidney disease patients on dialysis. Encouragingly, they show a positive association of MDS with LVH. 

However, I have some questions and concerns. 

Methods:

CAD was defined as, among other things, history of angina. While myocardial infarction, coronary intervention and bypass surgery are considered CAD, angina is only vaguely associated with the presence of CAD and should therefore not be considered as part of the CAD definition.

Echocardiography: References 30 and 31 refer to the same 2005 guidelines on ventricular quantification, which are out of date. A new guideline on chamber quantification was published in 2015 and should be considered. (DOI: 10.1016/j.echo.2014.10.003)

Were cardiac functional measures such as left ventricular ejection fraction or E/E' ratio available? It might be interesting to include functional measures in the analysis. Especially since diastolic dysfunction and heart failure with preserved ejection fraction (HFpEF) are common in the sample analysed.

Results:

In general: the title of tables and figures contains the statistical method. A title explaining the statement of the respective table/figure would be helpful. The method could be included in the title.

Since the association of Mediterranean diet (MDS) with LVH is being analysed, I would expect a table showing MDS by e.g. median/terile/quartile with clinical characteristics, cardiac structural/functional measures and LVH phenotypes. This would help me understand the sample and value of MDS. How do patients differ according to MDS?

There is a typo on page 6, line 209. I assume that r=-0.273 is meant. Also, I would not consider an r<0.5 to be a strong correlation. It is claimed that Table 4 shows a correlation with LVH. In fact, the table shows a correlation with left ventricular mass index (LVMi). It also states that all variables together explain 23% of the variability. However, the way the table is presented, I understand that PP explains 23% of the variability. Please clarify.

Is there really a difference between MDS scores according to LVH phenotype? There is a significant trend, but Figure 3 suggests that there is no significant difference. A statistical test would be helpful.

The groups in Table 6 should be defined in the Methods/Statistics section.

Is medication available to the patient? As heart failure medication blood pressure has an impact on LVH/LVMi, this could be considered in multivariate analyses.

In terms of limitations, I would only consider the use of LVMi as a weakness. More information could be obtained by including all standard measurements from a standard echocardiogram. An MRI would of course also be interesting.

Author Response

Dear Prof. Andres

Dear Prof. Hennerici,

Please find attached the revised version of our manuscript entitled “Adherence to the Mediterranean diet is associated with a more favourable left ventricular geometry in patients with End Stage Kidney Disease”.

We have responded to the reviewers’ comments and modified our manuscript accordingly.

We hope that the revised version of our manuscript now meets with your approval and will be accepted for publication in your highly esteemed journal.

Should you have any questions please do not hesitate to contact us.

On behalf of my co-authors,

Prof. K. Stylianou. Response to reviewers

RESPONSE TO REVIEWERS

Reviewer 1:

Comment 1:

 CAD was defined as, among other things, history of angina. While myocardial infarction, coronary intervention and bypass surgery are considered CAD, angina is only vaguely associated with the presence of CAD and should therefore not be considered as part of the CAD definition.

Response:

We thank the reviewer for this comment. As commented in the 2020 ESC Guidelines for the management of patients presenting with a non–ST segment elevation acute coronary syndrome, CKD-5D patients are less likely to receive an invasive strategy for diagnosing coronary artery disease[1]. This is the case concerning our dialysis patients. Nevertheless, it seems that there is no impact on the 1-year mortality among CKD-5D patients when an invasive strategy is chosen[1]. Unstable angina implies the presence of chronic myocardial injury[1]. In CKD-5D patients, there is an increased propensity for chronic myocardial injury; further, there is an increase in markers of myocardial injury that partially depict these alterations within their myocardium[2]. Therefore, we considered the presence of unstable angina as CAD analog, which might affect myocardial geometry in our patients.

We have made the appropriate corrections within the methods and discussion section as follows:

L.101-107: The rationale for including patients with angina pectoris as patients with CAD is the following: CKD-5D patients are less likely to receive an invasive strategy for diagnosing and treating CAD [29,30]. Furthermore, an increase in biomarkers of myocardial injury is less likely to be indicative of NSTEMI. The presence of chronic myocardial injury, which is common in CKD-5D patients, might as well result in an increase in myocardial injury biomarkers [29]. Therefore, the presence of angina pectoris was considered a CAD analog that might affect myocardial geometry.

L.328-329: Defining angina pectoris as a CAD analog might over-diagnose CAD within our patient cohort.

Comment 2:

Echocardiography: References 30 and 31 refer to the same 2005 guidelines on ventricular quantification, which are out of date. A new guideline on chamber quantification was published in 2015 and should be considered. (DOI: 10.1016/j.echo.2014.10.003)

Response:

We thank the reviewer for this comment. We acknowledge the presence of a new guideline on chamber quantification. However, all measurements were made according to 2005 Echocardiography guidelines. Given the cardiac ultrasound dependence on the operator, it is very likely to introduce a measurement bias since all measurements were made before 2015. We have commented on this in the discussion section and added the appropriate citation as follows:

  1. 329-333: Echocardiographic evaluation took place before the publication of the newest recommendation on chamber quantification in 2015[47]. In order to avoid a measurement/classification bias, given the operator variability in echocardiographic findings, we preserved the measurement definitions as described in the 2005 guideline on chamber quantification [33].

Comment 3:

Were cardiac functional measures such as left ventricular ejection fraction or E/E' ratio available? It might be interesting to include functional measures in the analysis. Especially since diastolic dysfunction and heart failure with preserved ejection fraction (HFpEF) is common in the sample analyzed.

Response:

We thank the reviewer for this comment. There are no data concerning E/E´ ratio. The proposal of the reviewer represents an excellent idea that we can apply in a future study.

Comment 4:

In general: the title of tables and figures contains the statistical method. A title explaining the statement of the respective table/figure would be helpful. The method could be included in the title.

Response:

Point well taken. When appropriate, we performed the correction proposed by the reviewer.  Specifically, L. 210 – 211, L. 226-227.

Comment 5:

Since the association of the Mediterranean diet (MDS) with LVH is being analysed, I would expect a table showing MDS by e.g. median/tertile/quartile with clinical characteristics, cardiac structural/functional measures and LVH phenotypes. This would help me understand the sample and value of MDS. How do patients differ according to MDS?

Response:  This comment helped us to further improve the manuscript. Table 5 shows the association of MDS (as a dichotomous variable) with clinical characteristics. However, as requested by the reviewer, we further analyzed the association of MDS quartiles with clinical characteristics, cardiac structural/functional measures and LVH phenotypes. We found an association of MDS quartiles with the presence of LVH or not (p for trend=0.005), with LV geometry pattern (p for trend =0.041), PVD (p for trend =0.048), stroke (p for trend =0.05) and need for b-blockers (p for trend =0.03). These data were included in a new table (Table 6) while previous Table 6 has now been renumbered as Table 7 in the revised manuscript.

This analysis based on the reviewer’s comment, gave interesting findings that made us add a new paragraph (3.7), as follows:

“ 3.7. MDS as a risk indicator for CVD

To provide further proof of the beneficial effect of adherence to MD, especially in the context of prevention of CVD, we examined the association of MDS categories (quartiles) with cardiac structure and LVH phenotypes, co-morbidities and antihypertensive drugs (Table 6). Participants with the highest levels of adherence to the MD (Q4) had a significantly lower risk of developing LVH (37.5% vs. 71.4% in Q1; p= 0.009) and abnormal LV geometry (eLVH: 18.8% in Q4 vs. 35.7% in Q1 and cLVH: 12.5% in Q4 vs. 32.1 % in Q1; p=0.020). In addition, patients in the highest quartile (Q4) had the lowest prevalence of PVD (18.8 % vs. 43.2% in Q1-Q3 combined; p=0.013) and stroke (3.1% vs. 18.9% in Q1-Q3 combined; p=0.030).  Finally, considering that a higher MDS was associated with the lowest rate of b-blockers use, greater adherence to MD may even reduce the demand for antihypertensive drugs, particularly b-blockers which are beneficial in the treatment of CVD.  Taken together all these data suggest that MDS could help identify CKD-5D patients at greater CV risk.

Table 6: Cardiac measurements, co-morbidities and antihypertensive drugs according to the quartiles of MDS

VARIABLES

Q1(≤22)

n=28

Q2 (23-24)

n =37

Q3 (25-26)

n=30

Q4(≥27)

n=32

P

for trend

Cardiac measures

A. LVH

0.005

absent

27.6 (8)

27.8 (10)

26.7 (8)

62.5 (20)

present

71.4 (20)

73.0 (27)

73.3 (22)

37.5 (12)

B. Geometry pattern

0.041

normal

24.1 (7)

19.4 (7)

16.7 (5)

34.4 (11)

cLVR

6.9 (2)

11.1 (4)

13.3 (4)

34.4 (11)

eLVH

35.7 (10)

37.8 (14)

43.3 (13)

18.8 (6)

cLVH

32.1 (9)

33.3 (12)

26.7 (8)

12.5 (4)

Comorbidities

PVD

39.3 (11)

51.4 (19)

36.7 (11)

18.8 (6)

0.048

CAD

14.3 (4)

32.4 (12)

36.7 (11)

28.1 (9)

0.256

STROKE

14.3 (4)

27 (10)

13.3 (4)

3.1 (1)

0.050

Anti- HTN drugs

ACEI/ARB

57.1 (16)

37.8 (14)

26.7 (8)

50 (16)

0.086

B-BLOCKERS

78.6 (22)

64.9 (24)

80 (24)

48.4 (15)

0.030

CCB

46.4 (13)

37.8 (14)

43.3 (13)

35.5 (11)

0.816

Comment 6:

There is a typo on page 6, line 209. I assume that r=-0.273 is meant. Also, I would not consider an r<0.5 to be a strong correlation. It is claimed that Table 4 shows a correlation with LVH. In fact, the table shows a correlation with left ventricular mass index (LVMi). It also states that all variables together explain 23% of the variability. However, the way the table is presented, I understand that PP explains 23% of the variability. Please clarify.

Response:

The assumption of the reviewer is correct. We have corrected the typographical error. The reviewer correctly pinpoints the fact that from a statistical point of view one might consider a correlation of 0.27 as a weak correlation. However, within the context of our study, which included a small number of patients, we don’t consider this correlation weak, given a statistical significance of 0.002. However, as suggested by the reviewer, we have omitted the word “strongly” from the text body so as to comply with the internationally accepted standards concerning correlation classification.

In the materials and methods section, we define LVH as an increase in LVMI. In order to avoid confusion, we have made the appropriate corrections within section 3.4.

Regarding Table 4: Pulse pressure is the last variable added to the multiple regression analysis model. The adjusted R2  in each row did not describe the individually contributed variability per variable, but the cumulative one. Hence, MDS confers a variability of 9% whereas PP confers a variability of 3.6% and the cumulative variability of all four variables is 23%. We have made the appropriate changes in Table 4, by adding individual and cumulative contributions, to clarify this issue.

Comment 7:

Is there really a difference between MDS scores according to LVH phenotype? There is a significant trend, but Figure 3 suggests that there is no significant difference. A statistical test would be helpful.

Response:

We thank the reviewer for this comment. MDS scores were calculated using multinomial regression analysis for their trend when compared with the normal group as described in the materials and methods section. Indeed, there was a significant trend for the whole group and statistically significant differences among the 3 hypertrophy groups. P values were added in the relevant figure.

Comment 8:

The groups in Table 6 should be defined in the Methods/Statistics section.

Response:

We have defined the groups in Table 6 (now table 7) in the Materials and methods section, and we further modified the text in the discussion section accordingly.

Comment 9:

Is medication available to the patient? As heart failure medication blood pressure has an impact on LVH/LVMi, this could be considered in multivariate analyses.

Response:

This is a very interesting suggestion. In Table 2 we describe the several medication classes for our patients: Most patients with LVH were on CCBs and beta–Blockers. There was no difference in ACEi/ARBs or diuretics use and the presence of LVH. Most patients receiving calcimimetics did not present with LVH (a statistically non-significant trend). Given the small number of patients, a given increase in the variables (each pharmacological category should be added) could perpetuate the multivariate analysis. We preferred to include PP as an indicator of arterial hypertension and its effect on LVMI[3].

Comment 10:

In terms of limitations, I would only consider the use of LVMi as a weakness. More information could be obtained by including all standard measurements from a standard echocardiogram. An MRI would of course also be interesting.

Response:

We agree with the reviewer. Unfortunately, cMRI data are not available.

Reviewer 2:

Comment:

The manuscript is well written and concise, however, there are few minor concerns - 

Abstract: Needs correction for “n=” in methods as Line 24; indicates 69 Men and 58 women, while Table 2: Line 177; Sex [(male=1/females=2 (n)] 77/50. 

Can authors provide the ethnicity of the patients available to include in the study? 

Minor editing required 

Response:

We thank the reviewer for her/his constructive criticism. We have corrected the proposed comments.

All patients were of Caucasian (Greek) ancestry. We have made the appropriate addition in the materials and methods section.

Reviewer 3:

Comment:

Please clarify how information on Mediterranean diet were provided to the patients; by dietician? How about those who did not have knowledge on Mediterranean diet?

Response:

We thank the reviewer for this comment. Information on the Mediterranean diet was provided by a dietician and a nephrologist. There weren’t any patients unaware of the Mediterranean diet.

Comment:

How did authors ascertain that patients really eat Mediterranean diet? questionnaires? 

Response:

The patients were interviewed personally (in person or by telephone call) by an experienced dietician (all by the same dietician). So, the bias of different perspectives of the dietician does not apply. The interview was based on Panagiotakos questionnaire of 11 food groups (alcohol included) that is attached as table 1. 

Comment:

How many patients were excluded for many reasons? Please provide flow diagram as figure.

Response: We thank the reviewer for this comment. A total of 540 dialysis patients were questioned. Of those 413 were excluded and 127 were included. An extra flow diagram was added as supplementary figure 1.

Comment:

Were those on Mediterranean diet on hemodialysis required lower potassium bath during dialysis?

Response:

We thank the reviewer for this comment. No difference between potassium baths was reported. Please keep in mind that this study was observational, not interventional.

Comment:

Rate of fluid removal should also be provided.

Response: This is a very interesting point for future studies. Since we included patients with an established dry weight, our study design does not allow us to make an extrapolation on fluid removal and the Mediterranean diet.

Comment:

How many patients refused Mediterranean diet?

Response: We thank the reviewer for this comment. Diet involves a spectrum of eating patterns. Panagiotakos questionnaire as presented in table 1 of the manuscript reflects this point. A score of 0 means zero compliance with the Mediterranean diet. None of our study participants had a score of 0. This is of importance since some patients include less Mediterranean diet components while others, the ones with higher MDS scores, have more Mediterranean diet components in their eating habits.

Literature:

  1. Collet, J.P.; Thiele, H.; Barbato, E.; Barthelemy, O.; Bauersachs, J.; Bhatt, D.L.; Dendale, P.; Dorobantu, M.; Edvardsen, T.; Folliguet, T.; et al. 2020 ESC Guidelines for the management of acute coronary syndromes in patients presenting without persistent ST-segment elevation. Eur Heart J 2021, 42, 1289-1367, doi:10.1093/eurheartj/ehaa575.
  2. Gunsolus, I.; Sandoval, Y.; Smith, S.W.; Sexter, A.; Schulz, K.; Herzog, C.A.; Apple, F.S. Renal Dysfunction Influences the Diagnostic and Prognostic Performance of High-Sensitivity Cardiac Troponin I. J Am Soc Nephrol 2018, 29, 636-643, doi:10.1681/ASN.2017030341.
  3. Jokiniitty, J.M.; Majahalme, S.K.; Kahonen, M.A.; Tuomisto, M.T.; Turjanmaa, V.M. Pulse pressure is the best predictor of future left ventricular mass and change in left ventricular mass: 10 years of follow-up. J Hypertens 2001, 19, 2047-2054, doi:10.1097/00004872-200111000-00016.

Reviewer 2 Report

The manuscript titled “Adherence to the Mediterranean diet is associated with a more favorable left ventricular geometry in patients with End Stage Kidney Disease” by Bacharki et al. have identified an association between adherence to Mediterranean diet and left ventricular hypertrophy (LVH) and cardiac geometry (in terms of LV mass index -LVMI and Relative wall thickness -RWT) particularly in chronic kidney disease patients on dialysis using multiple regression analysis. The study considered relevant parameters that may have confounding effect including sex, age, BMI, dialysis, comorbidities, smoking, medication and so on. Interestingly, the findings suggests for 18% lower chances of LVH with every 1-point greater MDS indicates relevance of nutritional intervention as a therapeutic measure in CKD patients.  

The manuscript is well written and concise, however there are few minor concerns - 

  1. Abstract: Needs correction for “n=” in methods as Line 24; indicates 69 Men and 58 women, while Table 2: Line 177; Sex [(male=1/females=2 (n)] 77/50. 

  1. Can authors provide the ethnicity of the patients available to include in the study? 

  2. Minor editing required 

Author Response

(The authors gave the same response as above.)

Reviewer 3 Report

1. Please clarify how information on Mediterranean diet were provided to the patients; by dietician? How about those who did not have knowledge on  Mediterranean diet?

2. How did authors ascertain that patients really eat  Mediterranean diet? questionnaires? 

3. How many patients were excluded for many reasons? Please provide flow diagram as figure.

4. Were those on  Mediterranean diet on hemodialysis required lower potassium bath during dialysis?

5. Rate of fluid removal should also be provided.

6. How many patients refused  Mediterranean diet?

Author Response

(The authors gave the same response as above.)

Round 2

Reviewer 1 Report

Dear Prof Stylianou and colleagues,

thank you for addressing all my comments so nicely.  Table 6 really helped me to understand the data. 

I just have 2 minor suggestion: 
You commented on Table 6 an - among others - stated that MDS is related to reduction in hypertensive medication which I completely agree with. However, I would be interested in how BMI - as a function of dietary diet - would be distributed across MDS quartiles.

As for the chamber quantification guideline, I did not realize when patients were recruited. However, I could not find recruitment time in the manuscript. Therefore, please add the years in which patients were recruited in the manuscript.

Thank you!

Thank you!

Author Response

Dear Prof. Andres

Dear Prof. Hennerici,

Please find attached the revised version of our manuscript entitled “Adherence to the Mediterranean diet is associated with a more favorable left ventricular geometry in patients with End Stage Kidney Disease”.

We have responded to the reviewers’ comments and modified our manuscript accordingly. All corrections are underlined and notified in yellow color.

We hope that the revised version of our manuscript now meets with your approval and will be accepted for publication in your highly esteemed journal.

RESPONSE TO REVIEWER

Comments

1) I just have 2 minor suggestions: 
You commented on Table 6 and - among others - stated that MDS is related to a reduction in hypertensive medication which I completely agree with. However, I would be interested in how BMI - as a function of dietary diet - would be distributed across MDS quartiles.

Response. We looked at BMI and the MD quartiles association. There was a trend for lower BMI in the higher MD quartile, but it didn’t reach statistical significance. We have added relevant information in paragraph 3.7 and Table 6, as follows: “Finally, BMI in the higher quartile (Q4) of MDS distribution (25.2±3.6 Kg/m2) was lower as compared to that of Q1-Q3 combined (26.6±5.0 Kg/m2), without however reaching statistical significance (p=0.151).”

2) As for the chamber quantification guideline, I did not realize when patients were recruited. However, I could not find recruitment time in the manuscript. Therefore, please add the years in which patients were recruited in the manuscript.

Response:

We added this information in paragraph 2.1 (study population): Patients’ recruitment into the study lasted from November 2019 to March 2020.

As stated in the weaknesses section we used the older chamber quantification guideline in order to maintain homogeneity between different hospitals.

22 Sep 2022

Should you have any questions please do not hesitate to contact us.

On behalf of my co-authors,

Prof. K. Stylianou.